# Children’s Community Nutrition Environment, Food and Drink Purchases and Consumption on Journeys between Home and School: A Wearable Camera Study

**DOI:** 10.3390/nu14101995

**Published:** 2022-05-10

**Authors:** Christina McKerchar, Ryan Gage, Moira Smith, Cameron Lacey, Gillian Abel, Cliona Ni Mhurchu, Louise Signal

**Affiliations:** 1Department of Population Health, University of Otago, Christchurch 8140, New Zealand; gillian.abel@otago.ac.nz; 2Health Promotion and Policy Research Unit, Department of Public Health, University of Otago, Wellington 6242, New Zealand; ryan.gage@otago.ac.nz (R.G.); moira.smith@otago.ac.nz (M.S.); louise.signal@otago.ac.nz (L.S.); 3Department of Māori and Indigenous Health Innovation, University of Otago, Christchurch 8140, New Zealand; cameron.lacey@otago.ac.nz; 4National Institute for Health Innovation, University of Auckland, Auckland 1142, New Zealand; c.nimhurchu@auckland.ac.nz

**Keywords:** food environment, food purchase, food consumption, travel mode, wearable cameras

## Abstract

Children’s community nutrition environments are an important contributor to childhood obesity rates worldwide. This study aimed to measure the type of food outlets on children’s journeys to or from school, children’s food purchasing and consumption, and to determine differences by ethnicity and socioeconomic status. In this New Zealand study, we analysed photographic images of the journey to or from school from a sample of 147 children aged 11–13 years who wore an Autographer camera which recorded images every 7 s. A total of 444 journeys to or from school were included in the analysis. Camera images captured food outlets in 48% of journeys that had a component of active travel and 20% of journeys by vehicle. Children who used active travel modes had greater odds of exposure to unhealthy food outlets than children who used motorised modes; odds ratio 4.2 (95% CI 1.2–14.4). There were 82 instances of food purchases recorded, 84.1% of which were for discretionary foods. Of the 73 food and drink consumption occasions, 94.5% were for discretionary food or drink. Children on their journeys to or from school are frequently exposed to unhealthy food outlets. Policy interventions are recommended to limit the availability of unhealthy food outlets on school routes.

## 1. Introduction

The rising rates of childhood obesity worldwide [1,2] have focused attention on the community nutrition environment, and its impact on children’s dietary behaviours and body weight. The community nutrition environment refers to the type, location and accessibility of food outlets within a geographic area [3]. While many studies have focused on the community environment around a child’s home [4], the community nutrition environment surrounding schools is also of interest [5,6]. In many countries, fast-food outlets or convenience stores tend to cluster around schools in urban areas at a greater frequency than if they were distributed evenly through a city in a way unrelated to the school’s location [7,8,9,10,11]. This pattern has exacerbated over time [12] and is often socioeconomically [13,14,15] or ethnically patterned [14,16,17,18].

There is mixed evidence of an association between the community nutrition environment surrounding schools and a child’s body weight or eating behaviours [19]. A systematic review that analysed 31 studies (up to 2019) found 14 studies that showed a direct association between proximity and density of food establishments, mainly fast-food outlets, convenience stores and grocery stores around schools, and overweight and obesity in children and adolescents [19]. For example, a US study found that 14–15-year-old adolescents were more likely to be overweight if they attended schools that had a convenience store within a 10 min walking distance [20]. However, 13 studies found no association [19]. There are many limitations to food environment research that may explain these heterogeneous findings including inaccurate measurement of the food environment through relying on databases, [21,22,23], only measuring one or two specific food outlets rather than the food environment in total [24] or estimating exposure rather than measuring actual exposure [24,25,26].

A child’s journey to or from school is important and often unaccounted for in studies [5]. Two studies found that the retail environment on the route to school had a small effect on increasing unhealthy food consumed [19] and BMI over a one-year period [20]. The travel mode to school should be accounted for as food environment characteristics might be more important for active travellers who walk or cycle to school than non-active travellers who travel by vehicle [16].

Wearable cameras have been used to reliably measure travel mode on the journey to school [27], as well as food purchase and consumption [28,29,30,31] and the context of eating episodes [31,32], including food consumed on transport journeys [33]. They enable accurate measurement of food consumption, especially of snacking episodes which are often under reported using traditional dietary recall methods [28]. They have also been used to describe children’s environments [34], food outlets [35], and to measure food purchasing and consumption behaviour to or from school [36]. Previous studies have usually examined one specific variable such as the travel mode on the journey to school [27] or food purchasing behaviour and consumption [36]. However, the relationship between modes of travel and food environments has not been previously explored using this methodology.

In this study, we analysed the journey to or from school (henceforth journey) among a sample of 147 children aged 11–13 years, using photographic images collected in the New Zealand (NZ) Kids’Cam study [37,38]. We aimed to:Measure the various types of food outlets to or from school that children are exposed to, taking into account the mode of transport used;Examine differences in exposure to food outlets by key sociodemographic characteristics (ethnicity and school decile);Determine food purchasing and food consumption on their journeys.

## 2. Materials and Methods

Kids’Cam NZ was a cross-sectional study, which ran from July 2014 to June 2015. A total of 168 randomly selected children aged 11–13 years old wore a wearable camera for 4 days (Thursday to Sunday). The camera took a 136° image of the children’s environments every 7 s producing approximately 1.3 million images. Children with a range of socio-economic and ethnic backgrounds were included within the sample, including a high proportion of indigenous Māori children, as well as Pacific children [39,40].

In NZ, schools are ranked on a decile system, which reflects the socioeconomic deprivation of the area from which the school draws its student roll. Low decile schools are in suburbs with high deprivation and high decile schools in more affluent areas. The sampling strategy was also designed to ensure students were represented from schools across the decile spectrum. To achieve this, the sampling frame included schools from across the Wellington region. In 2014, researchers obtained a list of all schools from the Wellington region whose roll included year 8 students. This list included the total numbers of year 8 children at the school as well as the total numbers by ethnicity. The list was then stratified into three groups based on decile rating. Low decile schools are those with a ranking of 1–3, medium decile schools had a ranking of 4–7, and high decile schools were those with a ranking of 8–10. This sampling was performed separately for each of the three decile groups as well as each of the three ethnic groups (Māori, Pacific, NZ European) [37]. For analysis schools were grouped into three tertiles: low (deciles 1–3), medium (deciles 4–7) and high (deciles 8–10). The study received ethical approval from the University of Otago Human Ethics Committee (Health) (13/220) to analyse the data for any topic of public health concern. The children were thus blinded to the aims of this specific study. Further details of the Kids’Cam study methods have been published previously [37,41].

### Coding and Data Analysis

Children who had image data showing their school journey were eligible for inclusion. This included images from Thursday and Friday morning and afternoons. Images for each participant were accessed using a password-protected hard drive. The coding excluded journeys with incomplete data.

A study protocol was developed and tested prior to coding. Two researchers (CM and RG) carried out a reliability test based on 67 selected photos and achieved 88% concurrence on outlet type. Food outlet definitions were tightened as a result. Both coders had 100% concurrence relating to food purchase and consumption occasions. The final study protocol ‘Journey to school’ is available at https://www.otago.ac.nz/heppru/research/index.html (accessed on 6 May 2022) [42].

One researcher (CM) coded all the images of the school journey. Coding began from the first image when a child left their home property in the morning until their arrival at their school grounds. Coding began again once a child had left the school grounds and finished on entry to their home gate. The start and finish times of each journey were recorded. The type of travel was also coded. This included active modes of transport including walking, scootering, and biking and motorised modes, private vehicle and public transport, including buses or trains. If a child used different travel modes for example if they walked to a bus stop and then continued on a bus, the separate modes of travel were each coded and this journey was classified as ‘mixed’.

Food outlets were identified through signage and other identifying features. Food outlet categories coded were as follows: convenience store; fast-food outlet; bakery; service station; café; ice-cream/gelato/yoghurt store; sweet shop; vending machine (non-core); vending machine (core); sushi shop; sandwich shop; medium supermarket; fruit and vegetable grocer; large supermarket; fresh food market; natural food store; juice bar; salad bar; mobile food vendor; other (core); and other (non-core). Food outlet category definitions were developed based on related research [38,43] and are detailed in the protocol. Vending machines were classified as either healthy (core) or discretionary (m-core) based on the classification of foods they contained using the WHO Regional Office for Europe Nutrient Profiling Model as for previous Kids’Cam studies [37]. In brief, core foods included vegetables and fruit, bread and cereals, milk, and meal and alternatives, and non-core foods were foods high in sugar, fat and salt including sugary drinks, confectionary and snack foods such as crisps [37].

The location and name of each food outlet was validated using Google street view [44,45]. Food outlets were usually only coded once. On rare occasions, if a child walked past an outlet and then after some time (30 min) returned to the outlet, this was coded again as a second food availability encounter. Figure 1 gives examples of food outlet images.

Food or drink purchase was coded if the images showed an item being purchased at a shop counter. If a participant purchased an item, this was coded as food purchase ‘participant’. If someone in the participant’s peer group or an adult who was present in the images, purchased an item, this was coded as food purchase ‘peer’. To code this, a peer had to be present in the series of images leading up to the food purchase, e.g., seen walking with the participant or travelling with them in a vehicle.

Food consumption was coded when a sequence of images showed a food or drink item being consumed. Foods and drinks were categorised into healthy (core) or discretionary (non-core) for analysis. This classification was based on the WHO nutrient profile model for marketing to children [46].

A Microsoft Excel spreadsheet was used for the coding and descriptive statistical analysis of the image data. Children’s demographic characteristics, transport mode, types of food outlets, purchase and consumption behaviours were summarised using descriptive statistics (counts and percentages for categorical data).

For analysis, food outlets were also categorised using the following classification system developed by Ferguson [47]. This was developed to allow comparison between studies monitoring the availability of healthy and unhealthy foods [48].

BMI healthy: fruit and vegetable grocer; large supermarket; natural food store; fresh-food market; juice bar; salad bar; vending machine (Core); Other (Core);BMI intermediate: sushi shop; sandwich shop; medium supermarket;BMI unhealthy: fast-food outlet; bakery; sweet shop; service station; ice-cream/gelato/yoghurt shop; convenience store; café, vending machine (non-core); mobile food vendor; and other (non-core).

To describe the exposure to food outlets by demographic variables, further statistical analyses were conducted in Stata/16. Due to the stratified sampling of schools and children, inverse sampling weights using Stata’s svy weights and associated weighting options were applied. This was performed so that the results were reflective of the Wellington population of children in this age-group and to ensure the inferential statistics (confidence intervals, *p*-values) were properly estimated to account for the survey sampling design. As multiple journeys were recorded for each child, analyses were also clustered by child. Logistic regression models were used to evaluate differences in exposure to BMI unhealthy or BMI healthy food outlets by travel mode, gender, ethnicity, and school tertile. The unadjusted counts for exposure to outlet type by demographic variable, are presented and then the adjusted proportions and confidence intervals.

## 3. Results

A total of 147 children or 87.5% of the Kids’Cam NZ participants collected image data that showed at least 1 journey to or from school. There were 444 journeys in total, a mean of 3.3 journeys per child. A total of 66 children (44.9%) collected data for 4 journeys during the data collection period (2 journeys × 2 school days). A total of 33 (22.4%) had three journeys, a further 33 (22.4%) had 2 journeys, and 15 children (10.2%) had data for only 1 journey over the 2 school days. The median journey time was 13.3 (6.3–28.6) min, which differed depending on travel mode, and length of the journey.

### 3.1. Demographic Characteristics

Table 1 shows the demographic characteristics of the sample. Of the 147 children, the mean age was 12.6, and the sample was ethnically and socioeconomically diverse. Over one third of the children (35.4%) were of Māori ethnicity, and one quarter (24.5%) were Pacific. Children of NZ European ethnicity constituted 40.1% of the sample.

The measure of socioeconomic deprivation used was the New Zealand Index of Socioeconomic Deprivation for Individuals (NZiDep) [49]. Approximately 31.3% of children sampled were from households with a high level of deprivation, and 65.3% were from households with a lower deprivation level. A further measure used was school decile. In this sample, 36.7% of children, attended low tertile schools (deciles 1–3), and conversely, 34.7% of children in this sample attended high tertile schools (decile 8–10). The reported BMI values are based on age- and sex-standardised cut-offs.

As outlined in Table 2, of 444 journeys analysed, active modes of transport were used during 212 (47.7%), motorised modes were used during 187 (42.1%), and 45 journeys (10.1%) were mixed modes of travel. Most journeys (60.8%) contained images of food outlets. Of the 257 journeys that included a component of active travel (mixed and active travel), 123 (47.9%) had image data for a food outlet, whereas 38 (20.3%) motorised journeys had images of a food outlet.

Convenience stores were the most common food outlet (20.5%), followed by fast-food outlets (12.8%) and supermarkets (12.2%). The odds of exposure to food outlets (as a % of journeys) tended to be higher for mixed journey modes than active modes and relatively rare for motorised journeys. For example, image data for one or more convenience stores was collected for 25.7% of journeys where children were actively travelling and 57.8% of mixed journeys compared to 4.3% of motorised modes. An exception was exposure to supermarkets, which was highest for modes that were motorised for all or part of the journey (10.1% of motorised modes and 44.4% of mixed modes). It was observed that there were some images collected from children sitting in parked cars outside supermarkets, or with an adult in a supermarket. This may be because a parent or caregiver driving may visit a supermarket on the way home to buy food. All other food outlets combined, such as cafes, or service stations or vending machines, made up 18% of exposures. Service stations appeared on 5.0% of journeys. Cafés featured at least once on 4.3% of journeys and bakeries on 3.8%. All other food outlets featured in image data on less than 2% of journeys.

### 3.2. Food Outlet Category, Travel Mode, and Demographic Characteristics

The food outlets were grouped into BMI healthy, BMI intermediate and BMI unhealthy categories for further analysis as illustrated in Table 3.

Table 3 shows the unadjusted counts for exposure to outlet type and the weighted proportion of children exposed to food outlets (with 95% confidence intervals), by journey mode and demographic variables. These have been estimated to account for sample weightings and clustering of multiple journeys by the same child.

The travel mode to school significantly influenced exposure to unhealthy food outlets. Children’s journeys by vehicle had proportionately less exposure to unhealthy food outlets 13.1% (95% CI 6.6 to 24.3) than for journeys with all (47.4% (95% CI 36.0 to 59.1)) or part of the journey travelling actively (82.2% (95% CI 59.5 to 93.6)).

All 147 children, regardless of the socio-economic profile of the school they attended, were more likely to have image data for BMI unhealthy food outlets than BMI healthy food outlets. For example, the 42 children who attended medium tertile schools, had image data for one or more BMI unhealthy food outlets on 40.7% (95% CI 28.4 to 54.3) of their journeys, whereas 8.5% (95% CI 3.6 to 19.0) of journeys had image data for BMI healthy food outlets.

Regardless of the child’s ethnicity, there was aa greater proportion of image data for BMI unhealthy food outlets than BMI healthy food outlets. In order to establish whether this difference persisted after adjusting for confounding factors especially mode of transport, logistic regression model analyses were performed and are presented in Table 4.

After adjusting for confounders, the most influential factor in exposure to BMI unhealthy food outlets was the mode of transport. Those children who undertook motorised journeys were significantly less likely to be exposed to unhealthy food outlets on their journey than those who used active transport. Children whose journey comprised mixed modes of transport were over four times more likely to be exposed to BMI unhealthy food outlets (4.2 95% CI 1.2 to 14.4) than children who used active modes of transport only.

Girls were more likely to be exposed to BMI unhealthy food outlets than boys (2.8 95% CI 1.3 to 5.7). Children who attended lower tertile schools had lower odds of exposure to BMI unhealthy food outlets 0.4 (95% CI 0.1 to 0.9) than children who attended medium tertile schools. Likewise, children who were of Pacific ethnicity were less likely to be exposed to BMI unhealthy food outlets (95% CI 0.2 to 0.9) than NZ European children. The estimate for Māori participants was compatible with that for Pacific participants; however, the statistical evidence for difference between Māori and NZ European (*p* = 0.11) is less robust than the evidence for Pacific than NZ European (*p* = 0.03).

For exposure to BMI healthy food outlets, again the most influential factor was transport mode, with children who had mixed modes of travel on their journeys being more likely (4.9 95% CI 1.5 to 15.8) to be exposed to BMI healthy food outlets than children using motorised modes. Māori children were 2.4 (95% CI 1.1 to 5.4) times more likely to be exposed to BMI healthy food outlets than NZ European children. There were differences in exposure by school tertile however these were not significant for most variables. Children attending lower tertile schools were less likely (0.5 95% CI 0.1–1.8) to be exposed to BMI healthy food outlets than those from medium tertile schools. Conversely, children from higher tertile schools were 2.5 (95% CI 0.8 to 7.8) times more likely than those attending medium tertile schools to be exposed to healthy food outlets.

### 3.3. Food Purchase

As shown in Table 5, the majority of purchases (84.1%) made on the journey to or from school were for discretionary foods, such as sugary drinks, ice-creams, and confectionery. Participants bought 60.9% of purchased items; however, 39.1% of purchases were by a peer of a similar age or by an adult they were with. The primary sources of discretionary food purchases were fast-food outlets and convenience stores followed by large supermarkets.

Food purchased at fast-food outlets included burgers and french-fries, and ice-creams. Frozen sugary drinks termed ‘slushies’ were also a popular item from fast-food outlets. Images of marketing for slushies were on advertisements on buses at the time of data collection, priced at NZ $1.00. See Figure 2 for examples. Confectionary, ice-creams, sugary drinks and potato chips made up the majority of purchases at convenience stores. On the two occasions that healthy foods were purchased at a convenience store, this was by an adult buying milk and bread.

Large supermarkets were the third most regular source of discretionary food items for children travelling to or from school. Many of the items purchased by participants at supermarkets included chocolate, potato crisps or sugary drinks, almost half (9/20) of which were then consumed by the participant as part of a discrete shop rather than as part of a more extensive grocery food shop. Figure 3 presents images of food purchases by participants in large supermarkets.

### 3.4. Consumption

Most (95.9%) of the food or drink items the children consumed were discretionary food and drink items from fast-food outlets, convenience stores and large supermarkets. In many cases, food was consumed while a participant was either walking or travelling in a car. There were also images of participants sharing food with peers while seated at fast-food outlets, or in shopping malls. Children occasionally consumed food with an adult (most likely a parent or caregiver) after school at fast-food outlets and in two instances at cafés. There were only two outlets where the food consumed was healthy; filled rolls from a sandwich outlet, and a salad and from a café.

Consumption (*n* = 20) of fast-food was higher than its purchase (*n* = 17) (Table 5). Children were occasionally given fast-food while seated in a car, perhaps if an adult had purchased food at a drive-through. In these cases, the purchase occasion was not seen in images if a child was sitting in a back seat. Figure 2 presents images of food consumption.

## 4. Discussion

Results from this study indicate that on the journey to or from school all children, regardless of ethnicity, gender, or school tertile, were more likely to be exposed to BMI unhealthy food outlets than BMI healthy food outlets. This was most significantly influenced by the mode of travel. Children who spent some of their time travelling actively had greater odds of exposure to a BMI unhealthy food outlet, especially a convenience store or fast-food outlet, than those who used only motorised transport modes.

Exposure to the type of food outlet was influenced in part by school tertile. Children who attended low tertile (more deprived) schools had less exposure to food outlets overall. Children from low tertile schools not only had less exposure to BMI unhealthy food outlets but also proportionately less exposure to BMI healthy food outlets on their journeys. This was tested through logistic regression modelling to account for confounding factors such as travel mode. Children from low tertile schools had lower odds of exposure to a healthy food outlet than children attending high tertile schools; however, the data observed are compatible with a wide range of potential differences between these groups as indicated by the wide confidence interval.

Children were also more likely to purchase and consume food if using an active mode of transport. This is consistent with studies that show that while active school travel is positively associated with a child’s activity levels, it has a limited impact on a child’s body weight due to the opportunities to purchase unhealthy foods enroute [50,51]. Active school travel may also increase the exposure of children to fast-food purchasing opportunities, differentially by ethnicity. In a US study, active school travel increased the exposure of Latino children in California to fast-food but had limited impact for other ethnicities [52]. In the current study, children of all ethnicities were exposed to a higher density of unhealthy food outlets compared to healthy food outlets. However, children of Pacific ethnicity had lower odds of exposure to BMI unhealthy food outlets at 0.4 (95% CI 0.1 to 0.9) odds ratio, and Māori children had 2.4 (95% CI 1.1–5.4) greater odds of exposure to BMI healthy food outlets. These results are unexpected and may reflect the relatively small number of schools (*n* = 16) within the overall sample. It is possible that the findings reflect the geographic location of a school (if it was close to a supermarket) which would increase the exposure to a BMI healthy outlet, and the number of children of different ethnicities who may have attended a specific school in this sample. In this study, most outlets categorised as BMI healthy were supermarkets, based on the classification system used [48]. In this classification system, it was acknowledged that supermarkets do stock unhealthy foods, yet they also have many healthy foods, which contributed to their classification as BMI healthy. Our findings, however, indicate that children on their own or with peers of their age (11–13) tended to purchase and then consume unhealthy foods and drinks from a supermarket. This finding is consistent with research that has found that supermarkets are a source of unhealthy foods for adolescents [53]. Food environment research needs to take into account how different age groups interact with food outlets.

Foodscapes vary and therefore need to be accounted for in their entirety [54]. In this study, convenience stores, fast-food outlets and supermarkets were the main food outlets, although children also purchased and consumed food from cafés, bakeries, ice-cream or yoghurt outlets, service stations and vending machines. This is consistent with research that has found that children and adolescents source foods from a wide range of outlets, including stores where food is not the primary item for sale [54,55].

Eating ‘on the go’ is associated with less healthy food choices for adolescents and young adults [33,56]. In a related study, unhealthy snacks were 15 times more likely to be consumed than healthy snacks when a child was in public spaces such as a food outlet [57]. Our findings show that most of the food purchased and consumed on school journeys was unhealthy. Food was often consumed in the car, or while a child was walking. Noticeably many children purchased and consumed food with their peers. Children of this age group (11–13 years old) have greater independence than young children but are not old enough to drive independently and tend not to be involved in meal preparation. Therefore, their food purchases tend to be limited to unhealthy snack foods [58].

Our findings indicate that programmes to encourage active transport to schools should also consider the food environment. Sustainability is a key driver of active transport initiatives due to associated reductions in greenhouse gas emissions [59], but the healthiness and sustainability of the food available also needs to be considered [60]. In NZ, researchers worked with a community to encourage an edible route to school that included community planting of fruit trees and fruits and vegetables in planter boxes, based on similar initiatives in England [61,62]. This is an example of a double or triple duty initiative that addresses the global syndemic of climate change, malnutrition and obesity [63].

Our findings also indicate that policy initiatives are required to improve the community nutrition environment. All children in this study, regardless of school tertile or ethnicity, were exposed to unhealthy food outlets on their journeys to or from school. A related study found that children’s environments in this age-group are relatively constrained to the immediate geographic area of their home and school, and 14% of their time not spent at school was spent in retail food outlets [34]. One possible policy initiative could be implementing healthy zones around schools to prevent unhealthy food sales and marketing [64]. In NZ, there are currently minimal mechanisms to control the proximity or density of food outlets in urban areas, and any changes to the current status-quo would require legislation [64].

While this research provides important new evidence on the community nutrition environment children encounter on their school journeys, it has some limitations. Children travelling by motorised transport modes were less likely to have image data for food outlets. This may be because the position of the camera did not enable an image to be collected. The probable scenario for this was when a child was driven to school in a private vehicle. The study design was cross-sectional; therefore, only associations can be drawn and causality cannot be inferred. The relatively small sample size limits the generalizability of the results. However, this study builds on previous findings [36] and demonstrates that wearable cameras can objectively measure the community nutrition environment for children. The use of wearable cameras reduces the need to use food diaries to record consumption, thereby reducing social desirability bias and participant burden [65]. Wearable cameras are useful in accounting for foods often undercounted using dietary recall methods [66], for example, foods consumed while travelling [27]. This methodology also limits the need to collect receipts to measure food purchase. Furthermore, it enables the context of an eating episode to be measured, for example, if a participant is consuming food on their own or with peers.

## 5. Conclusions

These results indicate that New Zealand children on their journeys to or from school are regularly exposed to unhealthy food outlets, especially convenience stores and fast-food outlets, and that they use supermarkets to purchase discretionary food. This exposure increases if a child is actively travelling. Policy and community measures are needed to limit the availability of unhealthy food outlets in children’s community settings and increase the opportunities for healthy eating.

## Figures and Tables

**Figure 1 nutrients-14-01995-f001:**
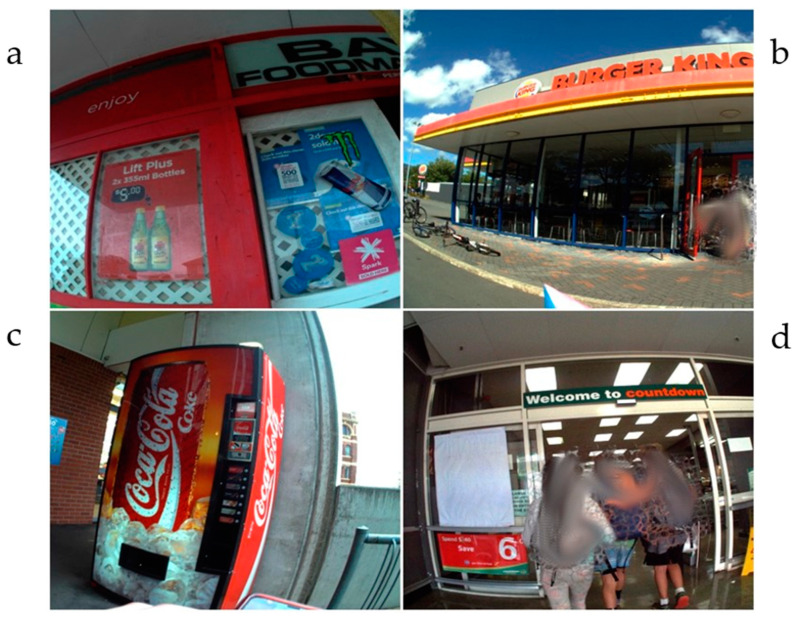
Food outlets. (**a**) convenience store; (**b**) fast-food outlet; (**c**) vending machine non-core; (**d**) supermarket.

**Figure 2 nutrients-14-01995-f002:**
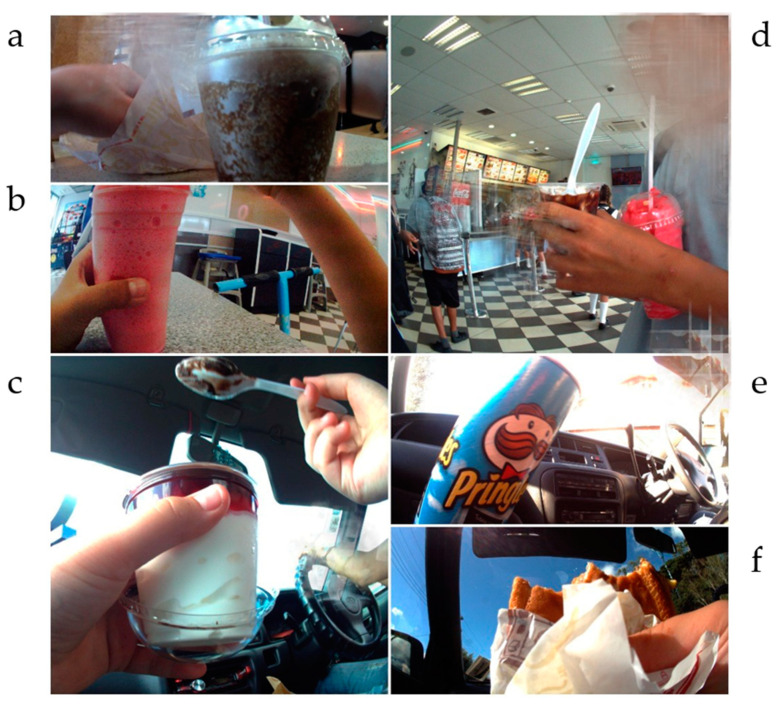
Purchase and consumption. (**a**) Frozen sugary drink at a fast-food outlet; (**b**) slushie at a fast-food outlet; (**c**) ice-cream sundae from fast-food outlet consumed while travelling in a car; (**d**) peer with frozen sugary drink and slushie at a fast-food outlet; (**e**) potato crisps purchased at a convenience store, consumed in a car; (**f**) hamburger from fast-food outlet consumed while travelling in a car.

**Figure 3 nutrients-14-01995-f003:**
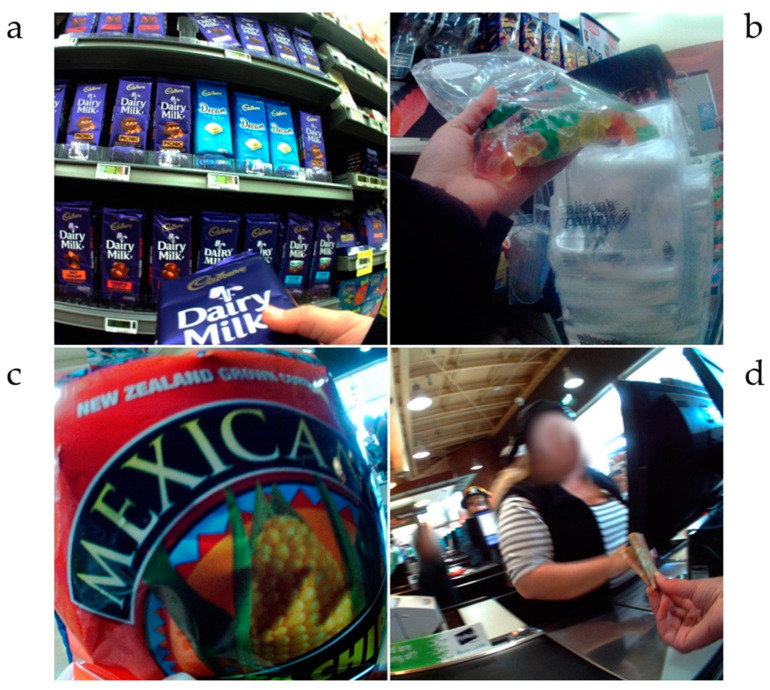
Food purchases in a supermarket. (**a**) chocolate; (**b**) confectionary from self-service pick and mix; (**c**) corn chips; (**d**) image of participant purchasing food at checkout.

**Table 1 nutrients-14-01995-t001:** Demographic characteristics of the sample.

Sample Characteristics		
Sociodemographic Variable and Group	*n*	%
Total	147	100
Gender		
Female	78	53.1
Male	69	46.9
Ethnicity		
NZ European	59	40.1
Māori	52	35.4
Pacific	36	24.5
Household socioeconomic deprivation *		
Lower (NZiDep 1–3)	96	65.3
Higher (NZiDep 4–5)	47	32.9
School tertile stratum		
Low (decile 1–3)	54	36.7
Medium (decile 4–7)	42	28.6
High (decile 8–10)	51	34.7
Age (years) **		
11	12	8.3
12	109	76.2
13	21	14.6
14	1	0.6
Mean (SD)	12.6 (0.5)	
BMI ***		
Not overweight (BMI = 16.0–24.9)	83	56.8
Overweight (BMI values ≥ 25.0)	63	43.2

* NZiDep missing for 3 participants (questionnaire not completed); ** age missing for three participants (questionnaire not completed); *** body mass index (BMI) missing for 1 participant as child declined to be measured.

**Table 2 nutrients-14-01995-t002:** Number of journeys in which there was exposure to one or more food outlet type by travel mode.

Food Outlet	Active Travel	%	Motorised Modes	%	Mixed	%	All	%
Bakery	7	3.3%	2	1.1%	8	17.8%	17	3.8%
Café	8	3.8%	2	1.1%	9	20.0%	19	4.3%
Convenience store	57	25.7%	8	4.3%	26	57.8%	91	20.5%
Fast-food outlet	30	14.2%	5	2.7%	22	48.9%	57	12.8%
Fresh-food market	1	0.5%	0	0%	0	0%	1	0.2%
Fruit and veg grocer	1	0.5%	0	0%	0	0%	1	0.2%
Ice-cream/gelato/yoghurt store	0	0%	0	0%	6	13.3%	6	1.4%
Juice bar	0	0%	0	0%	2	4.4%	2	0.5%
Large supermarket	19	9.0%	15	10.1%	20	44.4%	54	12.2%
Medium supermarket	2	0.9%	0	0%	5	11.1%	7	1.6%
Natural food store	0	0%	0	0%	3	6.7%	3	0.7%
Other miscellaneous	2	0.9%	0	0%	2	4.3%	4	0.9%
Sandwich shop	2	0.9%	1	0.5%	5	11.1%	8	1.8%
Service station	12	5.7%	7	3.7%	6	13.3%	25	5.6%
Sushi shop	0	0%	0	0%	4	8.9%	4	0.9%
Sweet shop	1	0.5%	0	0%	0	0%	1	0.2%
Vending machine core	0	0%	1	0.5%	0	0%	1	0.2%
Vending machine NC	1	0.5%	2	1.1%	5	11.1%	8	1.8%
Zero food outlet image	132	62.3%	149	79.7%	2	4.4%	161	36.2%
Total journeys	212	47.7%	187	42.1%	45	10.1%	444	

veg, vegetable.

**Table 3 nutrients-14-01995-t003:** Exposure to food outlet category, by travel mode and demographic characteristics.

		BMI U	% (95 CI)	BMI I	% (95 CI)	BMIH	% (95 CI)	Zero stores	% (95 CI)	Total/444
Journeys	Active	75	47.4 (36.0–59.1)	4	2.4 (0.6–8.8)	20	10.2 (4.8–20.2)	132	48.9 (37.3–60.6)	212
	Mixed	40	82.2 (59.5–93.6)	12	29.1 (12.5–54.0)	20	39.5 (22.6–59.4)	2	8.5 (1.3–39.5)	45
	Motorised	25	13.1 (6.6–24.3)	1	1.3 (0.2–8.9)	16	12.3 (6.6–21.7)	149	75.5 (64.9–83.8)	187
Gender	Male	63	26.5 (18.8–35.9)	7	4.2 (1.2–13.3)	26	14.4 (8.1–24.1)	141	65.4 (55.8–74.0)	214
	Female	77	48.2 (35.4–61.2)	10	7.3 (2.5–19.2)	30	16.2 (10.3–24.6)	142	44.1 (33.2–55.7)	230
Ethnicity	Māori	43	30.6 (22.4–40.3)	8	6.2 (2.9–12.7)	28	20.0 (12.7–30.1)	99	65.3 (55.5–73.9)	148
	Pacific	31	25.6 (15.9–38.2)	0	0	6	4.6 (2.1–9.7)	81	72.9 (60.6–82.5)	114
	NZE	66	41.2 (29.7–53.8)	9	6.8 (2.6–16.7)	22	16.4 (10.4–25.0)	103	48.7 (37.7–59.8)	182
School tertile	Low	33	22.1 (14.2–32.6)	1	0.6 (0.07–3.9)	6	4.1 (1.9–8.5)	122	76.6 (66.3–84.5)	157
	Med	48	40.7 (28.4–54.3)	5	4.3 (1.3–13.7)	11	8.5 (3.6–19.0)	70	57.7 (43.9–70.4)	120
	High	59	40.0 (27.7–53.7)	11	7.6 (2.9–18.4)	39	20.7 (13.8–29.9)	91	48.0 (36.3–59.9)	167

Abbreviations: BMI U, BMI unhealthy food outlet; BMI I, BMI intermediate food outlet; BMI H, BMI healthy food outlet; Zero stores, No image data for food outlets; NZE, NZ European ethnicity.

**Table 4 nutrients-14-01995-t004:** Odds ratios (with 95% confidence intervals) from logistic regression models for exposure to BMI unhealthy food outlets and BMI healthy outlets per journey adjusted for journey type, gender, ethnicity and school tertile stratum.

Demographic Factor	BMI Unhealthy Foods Outlets			BMI Healthy Food Outlets	
		Odds Ratio between Groups (95% CI)	*p* value	Odds Ratio between Groups (95% CI)	*p* value
Ethnicity					
Adjusted for school stratum, gender, journey type	NZ European	1.0		1.0	
	Māori	0.5 (0.3–1.2)	0.11	**2.4(1.1–5.4)**	**0.04**
	Pacific	0.4 (0.2–0.9)	0.03	0.5 (0.2–1.7)	0.28
School stratum					
Adjusted for ethnicity, gender, journey type	Low	0.4 (0.1–0.9)	0.02	0.5 (0.1–1.8)	0.3
	Medium	1.0		1.0	
	High	0.5 (0.3–1.2)	0.13	2.5 (0.8–7.8)	0.1
Gender					
Adjusted for ethnicity, school stratum journey type	Male	1.0		1.0	
	Female	**2.8 (1.3–5.7)**	**0.005**	0.9 (0.3–2.3)	0.8

Journey mode					
Adjusted for ethnicity, school stratum, gender	Motorised	**0.14 (0.06–0.33)**	**0.0001**	1.2 (0.4–3.4)	0.7
	Mixed	**4.2 (1.2–14.4)**	**0.02**	**4.9 (1.5–15.8)**	**0.007**
	Active	1.0			

Odds ratios were calculated accounting for the complex sampling design and were weighted to account for the oversampling of Māori and Pacific children. Odds ratios were mutually adjusted for all other variables in the model (ethnicity, school stratum, gender and journey type). Bolded text denotes statistically significant results *p* < 0.05.

**Table 5 nutrients-14-01995-t005:** Purchase and Consumption.

Food Outlets	Purchase					Consumption		
	Count	Participant	Peer	Non-Core	Core	Count	Non-Core	Core
Bakery	4	4	0	4	0	4	4	0
Café	6	2	4	4	2	6	5	1
Convenience store	18	16	2	16	2	16	16	0
Fast-food	17	8	9	17	0	20	20	0
Ice-cream/gelato/ Yoghurt store	3	2	1	3	0	3	3	0
Large supermarket	20	9	11	13	7	10	10	0
Medium supermarket	4	3	1	4	0	4	4	0
Other miscellaneous	1	1	0	1	0	1	1	0
Sandwich shop	3	0	3	1	2	3	1	2
Service station	3	3	0	3	0	3	3	0
Vending machine NC	3	2	1	3	0	3	3	0
Grand total	82	50	32	69	13	73	70	3

## Data Availability

The data presented in this study are available on request from the corresponding author. The data are not publicly available due to privacy and ethical restrictions.

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
