# Peer review of "Children’s Community Nutrition Environment, Food and Drink Purchases and Consumption on Journeys between Home and School: A Wearable Camera Study"

_nutrients, 2022, doi:10.3390/nu14101995_

Round 1

Reviewer 1 Report

This is a novel and interesting study using wearable cameras to assess the community food environment as children traveled to and from school. It is important to the field since it uses objective measures, and therefore it is important that it is a strongly written manuscript. In addition to the actual writing errors (e.g., many extra spaces between sentences, some duplicate words), the writing must be stronger and more mature. It is unacceptable for scientific publication as written.

Materials and Methods

Describe the sampling strategy

Specify the name of the protocol. There are four on the web page, and I couldn’t find the information I was looking for on any of them (how food outlets were coded).

Coding and data analysis 
Were each of the 1.3 million photos coded?

Results

In the Methods it stated the camera was worn from Thursday to Sunday. Are the authors only reporting the results of the two school days in the Results?

Table 1. Define NZiDep

Table 2 is never referred to in the text.

Table 3. Why are all of the significant ORs not bolded?

School tertile results for healthy outlets are described in the text as if they are significant, when they are not.

Figures 2 and 3 are in reverse order (Figure 3 is referred to before Figure 2)

Reviewer 2 Report

The paper uses data from an innovative study using photo images from children’s lives to analyze foods and beverages during commute to school. It’s a great study and the findings provide important insights on the sources and types of foods and beverages consumed by children, which can become the basis of policies regarding childhood nutrition and food environments. I believe the findings from this paper will be helpful to a wide range of audiences, from policy makers to school officials to nutrition researchers.

Just a few comments and questions:

1. BMI Healthy category includes Large Supermarket, Juice Bar, Vending Machine (Core), and Other

(Core). Could you please explain what “Core” means? Also, could you please briefly explain why you included supermarkets, juice bars, and vending machines in BMI Healthy as some products sold in those places are unhealthy/obesogenic?

2. You used BMI cutoffs of 25 but the participants are children/adolescents, so BMI z-scores may be more appropriate?

3. Given that the outcome (e.g., Table 4: the prevalence of exposure to BMI unhealthy stores) was high, could you explain why logistic regression models are appropriate to use with this common outcome?

4. I was curious about the BMI data of the children – did you analyze the association between the exposure to unhealthy/healthy food outlets and children’s BMI? What were the findings?
